# Differences in Tumour Aggressiveness Based on Molecular Subtype and Race Measured by [^18^F]FDG PET Metabolic Metrics in Patients with Invasive Carcinoma of the Breast

**DOI:** 10.3390/diagnostics13122059

**Published:** 2023-06-14

**Authors:** Sofiullah Abubakar, Stuart More, Naima Tag, Afusat Olabinjo, Ahmed Isah, Ismaheel Lawal

**Affiliations:** 1Department of Radiology and Nuclear Medicine, Sultan Qaboos Comprehensive Cancer Care and Research Center, Al-Khoud, Muscat 123, Oman; 2Department of Nuclear Medicine, Groote Schuur Hospital, University of Cape Town, Cape Town 7935, South Africa; 3Department of Radiology and Nuclear Medicine, Sultan Qaboos University Hospital, Al-Khoud, Muscat 123, Oman; 4Department of Obstetrics and Gynecology, Armed Forces Hospital, Al-Khoud, Muscat 123, Oman; 5Department of Nuclear Medicine, National Hospital, Abuja 90001, Nigeria; 6Department of Radiology and Imaging Sciences, Emory University, Atlanta, GA 30322, USA; ilawal@emory.edu

**Keywords:** [^18^F]FDG PET/CT, breast cancer, race, tumour aggressiveness, molecular subtype

## Abstract

Breast cancer in women of African descent tends to be more aggressive with poorer prognosis. This is irrespective of the molecular subtype. [^18^F]FDG PET/CT metrics correlate with breast cancer aggressiveness based on molecular subtype. This study investigated the differences in [^18^F]FDG PET/CT metrics of locally advanced invasive ductal carcinoma (IDC) among different racial groups and molecular subtypes. Qualitative and semiquantitative readings of [^18^F]FDG PET/CT acquired in women with locally advanced IDC were performed. Biodata including self-identified racial grouping and histopathological data of the primary breast cancer were retrieved. Statistical analysis for differences in SUVmax, MTV and TLG of the primary tumour and the presence of regional and distant metastases was conducted based on molecular subtype and race. The primary tumour SUVmax, MTV, TLG and the prevalence of distant metastases were significantly higher in Black patients compared with other races (*p* < 0.05). The primary tumour SUVmax and presence of distant metastases in the luminal subtype and the primary tumour SUVmax and TLG in the basal subtype were significantly higher in Black patients compared with other races (*p* < 0.05). The significantly higher PET parameters in Black patients with IDC in general and in those with luminal and basal carcinoma subtypes suggest a more aggressive disease phenotype in this race.

## 1. Introduction

Invasive ductal carcinoma (IDC), the dominant histological subtype of breast cancer [1], shows varying aggressiveness and prognosis depending on the molecular subtype. Generally, the basal subtype has a worse prognosis compared with the human epidermal receptor-2 (HER2)-enriched and luminal subtypes. Prognosis is best with the luminal subtype [2,3,4]. Poorer prognosis and higher mortality are seen in Black patients with breast cancer [5,6,7,8], and this has been attributed to socioeconomic factors as well as a more aggressive tumour biology. Late presentation, advanced stage disease, higher preference for breast conserving surgery and limited access to standard of care were some of the identified socioeconomic factors [8,9,10]. Factors related to tumour biology including higher incidence of aggressive molecular subtypes have also been implicated [11]. However, even within the same molecular subtype, a more aggressive disease with poorer prognosis is seen in Black patients, attributable to greater tumour heterogeneity [1] and a higher rate of genetic mutations driving aggressiveness [7,12,13]. The hazard of death from hormone receptor positive breast cancer was found to be at least four times higher in Black patients compared with their White counterparts, which is independent of stage, grade, or therapy initiation [7].

2-deoxy-2-[^18^F]fluoro-D-glucose ([^18^F]FDG) is a unique probe for molecular imaging with tumour uptake proportional to glucose utilisation by the tumour. The Warburg effect describes the increased aerobic glucose utilisation in cancer cells, which increases with the degree of de-differentiation and its aggressiveness. Hence, the more poorly differentiated a tumour is, the more the glucose utilisation and the more the [^18^F]FDG uptake [14,15,16,17]. The maximum standardised uptake value (SUV_max_), a semiquantitative [^18^F]FDG positron emission tomography (PET) metabolic metric of the most glucose-avid part of the tumour, is a molecular imaging marker of the degree of glucose utilisation which correlates with degree of de-differentiation and aggressiveness of a tumour. This has been demonstrated not just in breast cancer [18,19,20], but also in lung adenocarcinoma [14], lymphomas [17], colorectal cancer [21] and endometrial cancer [22]. The metabolic tumour volume (MTV), a measure of the metabolically active tumour size and the total lesion glycolysis (TLG), a measure of global glucose uptake in the tumour, have also been shown to correlate with tumour aggressiveness and are of prognostic significance in different cancers [23,24,25]. In breast cancer, high SUVmax, MTV and TLG predicted a high risk of adverse events or death [26].

The avidity of breast cancer for [^18^F]FDG is related to the histological type (IDC is more avid compared with lobular breast cancer), molecular type (triple negative breast cancer, TNBC, more avid than others) and complex oncogenic pathways activated within the tumour [27]. Upregulation of various oncogenic pathways leads to increased [^18^F]FDG uptake, while downregulation causes decreased uptake [27]. The various oncogenic pathways may partly explain differing tumour aggressiveness and prognosis in the same molecular subtype in different races; for example, prognosis is known to be worse in TNBC patients of African descent [12,13]. Within the same molecular subtype, Black patients were found to have greater intra-tumour heterogeneity and mutations driving a more aggressive tumour phenotype [12,13]. While there are studies on the correlation of [^18^F]FDG PET metabolic metrics and molecular subtypes in breast cancer [28], to our knowledge, there are none on the utility of the metrics as a marker of aggressiveness in breast cancer among different races. We investigated the differences in [^18^F]FDG PET metabolic metrics as a non-invasive tool to discriminate between the different molecular subtypes in locally advanced treatment for naïve IDC. We also investigated the differences in [^18^F]FDG PET metrics according to the races of these patients.

## 2. Materials and Methods

### 2.1. Patient Selection

This study is a retrospective review of [^18^F]FDG positron emission computed tomography with computed tomography (PET/CT) scans performed in breast cancer patients at the Groote Schuur Hospital, a provincial government-owned tertiary academic hospital of the University of Cape Town, South Africa. Patients referred to the Nuclear Medicine Department for initial staging of locally advanced histologically confirmed IDC of the breast with [^18^F]FDG PET/CT between January 2013 and December 2018 were identified from the departmental database. These patients had tumours greater than 5 cm at the widest dimension or tumours infiltrating the skin or chest wall. Histopathology was on core needle biopsy specimen in all patients. Male patients, patients who had removal of the primary tumour, neoadjuvant chemotherapy, recurrent breast cancer or histology other than IDC were excluded. All patients gave informed consent for the anonymous analysis and reporting of their data in a retrospective study. This study was approved by the Human Research Ethics Committee of the University of Cape Town (HREC REF: 061/2019).

### 2.2. [^18^F]FDG PET/CT Acquisition and Data Collection

Prior to [^18^F]FDG administration, patients fasted for 4 to 6 h while maintaining adequate hydration with plain water. Parenteral nutrition and glucose-containing infusions were stopped for 4 h. Diabetic patients with blood glucose above 11.1 mmol/L were rescheduled. Patients were kept warm on arrival at the department and given 20 mg oral propranolol 30 min prior to [^18^F]FDG administration (unless contraindicated) for brown fat [^18^F]FDG uptake suppression as per departmental protocol. A total of 2.8 MBq/kg (range 175 to 350 MBq) [^18^F]FDG was administered intravenously with an uptake time between 50 and 70 min. PET/CT images comprising non-contrast modulated low dose CT followed by PET emission images at 1.5 to 2.5 min per bed position, depending on the body mass index, were acquired on a dedicated PET/CT scanner (Philips Gemini TF PET/CT scanner) from the mid-thigh to the base of skull. Low-dose CT parameters were tube voltage 120 kV, current 30 to 100 mA (dose modulation) and slice thickness of 5 mm.

PET/CT images were reviewed on the Hermes hybrid viewer—HERMES Gold 3™ software (HERMES, Stockholm, Sweden). Volume of interest was generated using iso-contouring and a fixed relative threshold of 41% [29] to ensure uniformity of primary tumour segmentation. The SUVmax, MTV and TLG of the primary breast lesion, regional lymph nodes and distant metastases were recorded. For each patient, the histopathology report was reviewed and hormone receptor status was retrieved. Using the hormone receptor and HER-2 expressions, patients were sub-grouped as luminal, HER2-enriched and basal molecular subtypes [30]. Patients with hormone receptor positivity irrespective of HER2 status were grouped as luminal, those who were hormone receptor negative and HER2 positive were grouped as HER2-Enriched and those who were negative for hormone receptors and HER2 receptors were grouped as basal subtype. The biodata including the self-identified race of the patients were recorded. Race was categorised as Black, White and mixed ancestry. Mixed ancestry patients refer to the South African mixed-race community with more than one of European, African or Asian ancestry.

### 2.3. Statistical Analysis

Statistical analysis was performed with IBM SPSS statistics version 27. Continuous variables with normal distribution were analysed with ANOVA and two sample *T*-tests. Continuous variables that were not normally distributed were analysed with Kruskal–Wallis and Mann–Whitney U-tests. Categorical variables were subjected to a Chi square test. Statistical significance was set at *p* ≤ 0.05.

Analysis was performed to check for differences based on race in SUVmax, MTV and TLG of the primary tumour as well as the presence of regional or distant metastasis. Differences based on molecular subtype and race in the above parameters within each molecular subtype were also analysed.

## 3. Results

A total of 127 female patients with locally advanced breast cancer who had a staging [^18^F]FDG PET/CT scan were included in the study. Eighty-one (63.8%) of the patients were mixed ancestry, forty (31.5%) were Black and six (4.7%) were White. Their mean age was 55.5 years (standard deviation = 12.7). All patients had locally advanced breast cancer with a documented clinical stage of cT3N1 constituting 25%, cT4N0 18%, cT4N1 37% and cT4N2 20%.

On analysis by race (Table 1), Black patients were younger, with a mean age of 49.6 years, compared with the mixed ancestry (52.1 years) and White (55.8 years) patients, although this difference did not reach statistical significance (*p* = 0.425). There was a significant difference in primary tumour SUVmax (*p* = 0.004), MTV (*p* = 0.041) and TLG (*p* = 0.005) among the races, with Black patients having the highest values followed by mixed ancestry and White patients. One hundred and ten patients (86.6%) had regional metastases on [^18^F]FDG PET/CT with no significant difference among the races (*p* = 0.204). Distant metastases were seen in 61 patients (48%) and their presence was significantly higher in Black patients (70%), compared with mixed ancestry (39.5%) and White (16.7%) patients (*p* = 0.002).

In 119 patients, sufficient information on histopathology was available to group them into molecular subtypes. Of these 119 (Table 2), 80 patients (67.2%) had luminal, 20 (16.8%) had HER2-enriched and 19 (16%) had basal cancer subtypes. There was no significant difference in the age distribution, although patients in the basal group tended to be younger. The primary tumour SUVmax, MTV and TLG were significantly higher in the basal subgroup. There was no significant difference in the presence of regional or distant metastases among the three subgroups.

Within each molecular subtype, we compared the age of patients, [^18^F]FDG PET-derived metrics and the occurrence of regional or distant metastases between the different races (Table 3). We found no significant difference in age distribution among the races. However, the primary tumour SUVmax and the presence of distant metastases were significantly higher in the Black patients compared with the other races in the luminal subtype. There was no significant difference in the MTV, TLG and presence of regional metastases in the luminal subtype. In the HER2-enriched subgroup (Table 3), we found no significant difference in the primary tumour SUVmax, MTV, TLG and presence of regional and distant metastasis among the different races.

Due to the few numbers of White patients in our cohort, we excluded them and compared the mean age of patients, the median values of the [^18^F]FDG PET-derived metrics and the occurrence of regional or distant metastases between Black and mixed ancestry patients (Table 4). The results of the comparison were largely similar to the results obtained when White patients were included in the analysis, as shown in Table 3. There were no White patients with the basal molecular tumour subtype. A comparison of mixed ancestry versus Black patients with the basal tumour subtype showed significantly higher primary tumour TLG among Black patients compared with mixed ancestry patients (Table 4). Additionally, the primary tumour SUVmax was higher among Black patients compared with those of mixed ancestry, with a *p*-value tending toward statistical significance (*p* = 0.057). Within each subgroup, we did not find a significant difference in the proportion of patients based on race (Table 3 and Table 4). Figure 1 is a representative image of the findings.

## 4. Discussion

Breast cancer is a heterogeneous disease with several tumour- and patient-related factors affecting the aggressiveness and prognosis. The histological type and molecular subtype, in addition to guiding treatment strategy, are major determinants of aggressiveness and prognosis. The effect of race on aggressiveness and prognosis has been highlighted and is related to characteristics at presentation such as advanced stage at presentation, access to care and type of treatment received, but also to differences in tumour biology irrespective of presentation [12,13,31,32]. With the known correlation of intensity of [^18^F]FDG uptake with tumour aggressiveness in [^18^F]FDG-avid tumours [17,21,33], we investigated the differences in semiquantitative [^18^F]FDG parameters in locally advanced invasive ductal carcinoma based on race and molecular subtypes. We found that in locally advanced IDC as a whole, irrespective of molecular subtype, Black patients had significantly higher primary tumour SUVmax, MTV, TLG and presence of distant metastases compared with mixed ancestry and White patients. Within each molecular subtype, irrespective of race, we found the primary tumour SUVmax, MTV and TLG were significantly higher in the basal subtype, followed by the HER2-enriched subtype and least in the luminal subtype. Based on race, within each molecular subtype, Black patients had significantly higher SUVmax and presence of distant metastases in the luminal subtype and significantly higher TLG in the basal subtype. There was no significant difference in the [^18^F]FDG PET metrics or in the presence of regional or distant metastasis in the HER2-enriched subgroup in the different races.

The setting of our study was a tertiary government facility in Cape Town that provides government-funded or subsidised healthcare to the general population. The more affluent population with private health insurance can access care in private health facilities. This partly explains the differences in the number of patients in the different race groups, with the majority (63.8%) being of mixed ancestry. A higher age standardised incidence rate of breast cancer in the mixed ancestry compared with the Black population in South Africa [34] also explains the finding of our study. Younger patients tend to have more aggressive disease, especially the basal subtype [35], and in this study, Black patients tended to be younger in IDC as a whole and in the subtypes, although this was not significant. Younger age approached significance (*p* = 0.078) in the basal subtype compared with the other subtypes.

Features of aggressiveness in malignancies include short doubling time with fast growth rate and larger tumours and high potential for regional and distant metastases, all contributing to poorer prognosis. In IDC as a whole, the breast tumour was significantly larger (MTV) in Black patients compared with their mixed ancestry and White counterparts. The SUVmax and TLG of the primary tumour were also significantly higher. These higher [^18^F]FDG PET metrics (SUVmax, MTV and TLG) of the primary tumour support a faster growth rate and a more aggressive primary [14,15,28,36]. The aggressiveness of the primary breast cancer is corroborated by significantly higher distant metastatic disease in Black patients in this study (Figure 1).

A more aggressive locally advanced IDC may be explained by the preponderance of more aggressive molecular subtype by race. Further analysis was therefore conducted based on molecular subtype. In keeping with known aggressiveness based on molecular subtype [2,3,4,37], with luminal being the least aggressive, basal the most aggressive and HER2-enriched in between, SUVmax, MTV and TLG of the primary tumour were highest in basal, followed by HER2-enriched and least in the luminal subtype. These differences in quantitative [^18^F]FDG parameters were significant and in line with prior publications [15,18,19].

In the luminal subgroup, the Black patients demonstrated significantly higher SUVmax and the presence of distant metastases, suggesting a more aggressive disease. A study limitation is the inability to further classify into luminal A or the more aggressive luminal B^35^ due to insufficient information on histopathology. However, higher intra-tumour heterogeneity with increased genetic mutation driving a more aggressive disease was demonstrated in the luminal group as a whole in Black patients [18,38] and we propose that this is likely reflected by the significantly higher SUVmax and presence of distant metastases. The HER2-enriched subgroup did not show significant difference in [^18^F]FDG parameters and the presence of regional or distant metastases in the different races. In the basal subgroup, significantly higher TLG in the primary tumour and primary tumour SUVmax that approached significance in Black patients were likely due to known more aggressive disease in this group [13,39].

Poorer prognosis and higher mortality in women of African descent with breast ca [5,8,9,40] are related not only to socioeconomic status and access to care [9,41,42], but also to differences in tumour biology, intra-tumour heterogeneity and genetic mutations which drive a more aggressive disease [13,39]. The various genetic mutations and oncogenic pathways influence [^18^F]FDG uptake in these tumours and may be surrogates for aggressiveness [27]. We were able to demonstrate significantly higher SUVmax, MTV and TLG in the primary tumours that is in keeping with known aggressiveness of the molecular subtypes of breast cancer. In addition, this study showed significantly higher SUVmax, MTV and TLG of the primary tumour and likewise for the presence of distant metastases, which were all suggestive of a more aggressive disease in Black patients with locally advanced IDC. However, within the molecular subgroups, we were only able to show significantly higher [^18^F]FDG metrics in Black patients in the luminal (SUVmax and presence of distant metastasis) and basal (TLG) sub groups. To our knowledge, this is the first report of these findings on [^18^F]FDG PET/CT for locally advanced IDC based on race. It will be insightful to see how these differences in primary tumour PET-derived metrics, regional and distant metastases and the molecular phenotypes between the difference races impact treatment outcomes and patient survival. Differences in treatment regimen given to the patients precluded the inclusion of survival data in this work.

One of the limitations of this study is the utilisation of histopathology results from core needle biopsy and the non-availability of surgical specimen histopathology, which is expected to be more accurate. Histopathology on surgical specimens were not available in many patients included in the study as they did not have surgery due to advanced disease. The reported agreement between core needle biopsy and surgical specimen histopathology is, however, very good [43], and it is unlikely that this limitation had a significant impact on the outcome of the study. An [^18^F]FDG uptake time of 50 to 70 min is another potential limitation, as this may affect the SUV metrics. The uptake time is similar to recommendations by the standard guidelines, including the EANM guideline. Fewer White patients is also a limitation, and we conducted a reanalysis on account of this excluding the White patients and the results, as detailed in Section 3, are largely similar.

The anatomic size of the primary tumour was not included in the data or analysis. However, we ensured uniformity in disease stage by only recruiting patients with locally advanced disease. By this inclusion criteria, it means that all the patients recruited into this study had a primary tumour greater than 5 cm in their wider dimension or the tumour had infiltrated the skin or the chest wall. While larger tumour portends more advanced/aggressive disease, one potential downside of an overreliance on primary tumour size is the possibility of necrosis at very large tumour size, such that the measured tumour size does not represent just the viable tumour bulk but also areas of fibrosis and necrosis. This may explain why in the TNM staging of breast cancer, size of the tumour is not factored into staging beyond a diameter of 5 cm. In this study, rather than using primary tumour size to quantify the bulk of the primary tumour volume, we instead used metabolic tumour volume (MTV), which only accounts for the viable portion of the tumour and is superior to anatomic size in predicting disease aggressiveness and treatment outcome [44]. The higher incidence of distant metastases in Black patients is more likely due to a more aggressive primary rather than size of the primary or late presentation. Sopik et al. [45] showed little correlation between breast cancer tumour size and presence of distant metastases in small (less than 10 mm) and very large (>60 mm) tumours and non-linear correlation in tumours between 10 and 60 mm.

## 5. Conclusions

Black patients showed the highest [^18^F]FDG PET-derived metabolic metrics and the occurrence of distant metastases compared with White patients and patients of mixed ancestry. Basal molecular subtype of invasive ductal breast carcinoma was significantly associated with the highest [^18^F]FDG PET-derived metabolic metrics of primary tumour compared with other molecular subtypes. Taken together, tumour biology in Black patients appear to be more aggressive, and this supports prior findings that poorer prognosis and higher mortality in Black patients is not only related to socioeconomic factors, tumour characteristics at presentation and type of treatment received, but also to a more heterogeneous and aggressive disease biology.

## Figures and Tables

**Figure 1 diagnostics-13-02059-f001:**
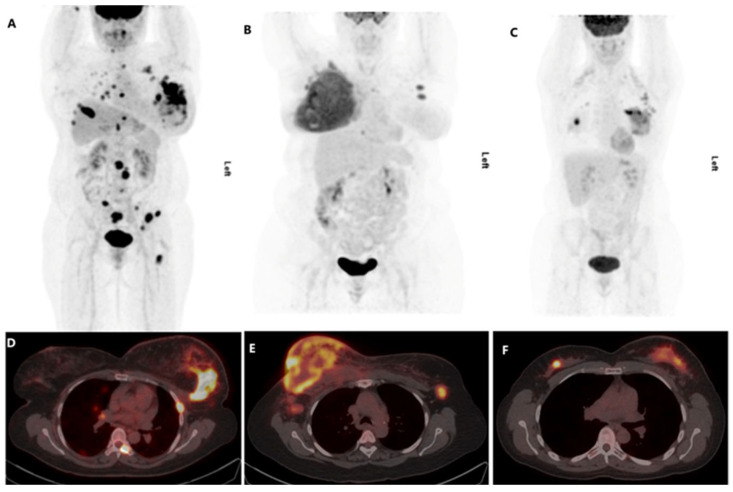
Representative maximum intensity projection and fused axial slices of the primary tumour. (**A**,**D**)—Black patient; (**B**,**E**)—Patient of mixed ancestry; (**C**,**F**)—White patient.

**Table 1 diagnostics-13-02059-t001:** Analysis by race in all patients with locally advanced IDC.

	Total	Mixed Ancestry	Black	White	*p*-Value
Number (%)	127 (100%)	81 (63.8%)	40 (31.5%)	6 (4.7%)	
Mean Age (SD), years	51.50 (12.78)	52.1 (12.45)	49.63 (13.51)	55.83 (12.58)	0.425 ^A^
Median SUVmax-PT	11.5	11.1	13.80	6.30	0.004 ^K^
Median MTV-PT	37.3	30	52.05	14.8	0.041 ^K^
Median TLG-PT	235.0	229.3	304.95	40.75	0.005 ^K^
Patients with regional metastases, *n* (%)	110 (86.6%)	67 (82.7%)	37 (92.5%)	6 (100%)	0.204 ^C^
Patients with distant metastases, *n* (%)	61 (48%)	32 (39.5%)	28 (70%)	1 (16.7%)	0.002 ^C^

SD: Standard Deviation; ^A^: Analysis of Variance; ^K^: Kruskal–Wallis; ^C^: Chi-square; SUVmax-PT: maximum Standardised Uptake Value of the Primary Tumour; MTV-PT: Metabolic Tumour Volume of the Primary Tumour; TLG-PT: Total Lesion Glycolysis of the Primary Tumour.

**Table 2 diagnostics-13-02059-t002:** Analysis by molecular subtype.

	Luminal	HER2-Enriched	Basal	*p*-Value
Number (%)	80 (67.2%)	20 (16.8%)	19 (16%)	
Mean Age (SD), years	51.8(12.6)	54.5 (13.9)	45.7 (10.9)	0.078 ^A^
Median SUVmax-PT	10.3	13	20.3	<0.001 ^K^
Median MTV-PT	24.45	62.8	119.6	<0.001 ^K^
Median TLG-PT	162.1	557.45	1384.6	<0.001 ^K^
Patients with regional metastases, *n* (%)	66 (82.5%)	20 (100%)	16 (84.2%)	0.376 ^C^
Patients with distant metastases, *n* (%)	36 (45%)	13 (65%)	9 (47.4%)	0.275 ^C^

SD: Standard Deviation; ^A^: Analysis of Variance; ^K^: Kruskal–Wallis; ^C^: Chi-square; SUVmax-PT: maximum Standardised Uptake Value of the Primary Tumour; MTV-PT: Metabolic Tumour Volume of the Primary Tumour; TLG-PT: Total Lesion Glycolysis of the Primary Tumour; HER2: Human Epidermal Receptor 2.

**Table 3 diagnostics-13-02059-t003:** Analysis based on molecular subtype and race.

Luminal				
	Mixed Ancestry	Black	White	*p*-Value
N/T (%) *	53/76 (69.7%)	23/37 (62.2%)	4/6 (66.7%)	0.723 ^C^
Mean Age (SD), years	53.4 (12.9)	47.9 (11.7)	53.5 (12.4)	0.216 ^A^
Median SUVmax-PT	9.7	12.1	4.8	0.012 ^K^
Median MTV-PT	22.5	27.9	27.8	0.411 ^K^
Median TLG-PT	151.1	189.7	77.05	0.132 ^K^
Patients with regional metastases, *n* (%)	41 (77.4%)	21 (91.3%)	4 (100%)	0.340 ^C^
Patients with distant metastases, *n* (%)	19 (35.8%)	17 (73.9%)	0 (0%)	0.006 ^C^
**HER2-Enriched**				
N/T (%) *	11/76 (14.5%)	7/37 (18.9%)	2/6 (33.3%)	0.452 ^C^
Mean Age (SD), years	54.2(13.4)	53.3 (16.0)	60.5 (16.3)	0.823 ^A^
Median SUVmax-PT	16.8	11.9	9.3	0.394 ^K^
Median MTV-PT	86.6	57.9	4.2	0.075 ^K^
Median TLG-PT	771	523.5	20.75	0.071 ^K^
Patients with regional metastases, *n* (%)	11 (100%)	7 (100%)	2 (100%)	NA
Patients with distant metastases, *n* (%)	6 (54.5%)	6 (85.7%)	1 (50%)	0.359 ^C^

*: Number in racial group with molecular subtype over total number in the racial group (percentage); SD: Standard Deviation; ^A^: Analysis of Variance; ^K^: Kruskal–Wallis; ^C^: Chi-square; SUVmax-PT: maximum Standardised Uptake Value of the Primary Tumour; MTV-PT: Metabolic Tumour Volume of the Primary Tumour; TLG-PT: Total Lesion Glycolysis of the Primary Tumour; NA: Not applicable; HER2: Human Epidermal Receptor 2.

**Table 4 diagnostics-13-02059-t004:** Analysis based on molecular subtype and race with white patients excluded.

Luminal			
	Mixed Ancestry	Black	*p*-Value
N/T (%) *	53/76 (69.7%)	23/37 (62.2%)	0.421 ^C^
Mean Age (SD)	53.4 (12.9)	47.9 (11.7)	0.085 ^T^
Median SUVmax-PT	9.7	12.1	0.017 ^M^
Median MTV-PT	22.5	27.9	0.182 ^M^
Median TLG-PT	151.1	189.7	0.081 ^M^
Patients with regional metastases, *n* (%)	41 (77.4%)	21 (91.3%)	0.149 ^C^
Patients with distant metastases, *n* (%)	19 (35.8%)	17 (73.9%)	0.002 ^C^
**HER2-Enriched**			
N/T (%) *	11/76 (14.5%)	7/37 (18.9%)	0.544 ^C^
Mean Age (SD)	54.2 (13.4)	53.3 (16.0)	0.899 ^T^
Median SUVmax-PT	16.8	11.9	0.650 ^M^
Median MTV-PT	86.6	57.9	0.860 ^M^
Median TLG-PT	771	523.5	0.724 ^M^
Patients with regional metastases, *n* (%)	11 (100%)	7 (100%)	
Patients with distant metastases, *n* (%)	6 (54.5%)	6 (85.7%)	0.171 ^C^
**Basal**			
N/T (%)	12/76 (15.8%)	7/37 (18.9%)	0.676 ^C^
Mean Age (SD), years	46.83 (10.62)	43.71 (12.06)	0.564 ^T^
Median SUVmax-PT	16.55	21.2	0.057 ^M^
Median MTV-PT	92.85	131.70	0.100 ^M^
Median TLG-PT	634.59	2842.5	0.007 ^M^
Patients with regional metastases, *n* (%)	10 (83.3%)	6 (85.7%)	0.890 ^C^
Patients with distant metastases, *n* (%)	6 (50%)	3 (42.9%)	0.763 ^C^

*: Number in racial group with molecular subtype over total number in the racial group (percentage); SD: Standard Deviation; ^T^: Two sample *T*-test; ^M^: Mann–Whitney U; ^C^: Chi-square; SUVmax-PT: maximum Standardised Uptake Value of the Primary Tumour; MTV-PT: Metabolic Tumour Volume of the Primary Tumour; TLG-PT: Total Lesion Glycolysis of the Primary Tumour; HER2: Human Epidermal Receptor 2.

## Data Availability

The data presented in this study are available on request from the corresponding author.

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
