# Peer review of "Differences in Tumour Aggressiveness Based on Molecular Subtype and Race Measured by [18F]FDG PET Metabolic Metrics in Patients with Invasive Carcinoma of the Breast"

_diagnostics, 2023, doi:10.3390/diagnostics13122059_

Round 1

Reviewer 1 Report

Very interesting subject and well written paper.

A have a two questions:

1.     I would like to know if the black patients presented with a higher stage or size that could also influence the PET metabolic parameters. Could you add this in the tables?

2.     Also is race an independent factor for aggressiveness, taking into account age, stage, histology and size? A multivariate analysis would be probably needed to address this issue.

Reviewer 2 Report

The authors designed a manuscript good quality, presenting an interesting and still underresearched subject regarding breast cancer and how its manifestation differs between races, which could provide valuable information on how we can better and more rapidly manage these type of patients. I still have some suggestions for some minor changes, that I believe would enhance even more the quality of the manuscript:

  1. In the description of the patient selection process, I suggest adding the methods used to obtain the histopathological exam (surgery, biopsy etc), as well as the number of patients who had each method performed. I believe that this may affect the results a bit, as a specimen obtained through biopsy might not produce results as accurate as the one obtained through surgery, so this should be also specified in the study limitations.
  2. I recommend adding a special paragraph regarding the study limitations at the end of the Discussion section, as there are several: the few White patients included in the study, the specimen obtaining method for the histopathological exam, the variable time interval between the injection time of F18-FDG and the beginning of the PET/CT (as the authors mention that F18-FDG-PET was acquired 50 to 70 min postinjection and it is well known that the F18-FDG decay in time may affect the PET parameter measurements) etc.
  3. I suggest performing an English language check, as there are some grammatical errors throughout the manuscript, such as in line 68 Black patients were found have greater”, line 222 “the breast primary” etc.

Reviewer 3 Report

Thank you very much for the opportunity to review the study performed by Abubakar et al. which has an aim to assess the differences in Tumor Aggressiveness Based on Molecular Subtype and Race Measured by [18F]FDG PET metabolic metrics in Patients with Invasive Carcinoma of the Breast. The subject raised by the authors is important in the field of lymphoma patients, however some changes are required before accepting.

1.       Please check the whole manuscript for the linguistic and stylistic errors as well as double space or no space, for example (but not limited to) in the whole Introduction section there is no space between last word and the references.

2.       Material and Methods section: please explain what “half body images” means.

3.       Material and Methods section: please add information about the parameters used for low-dose CT imaging: slice thickness, kV, mAs etc.

4.       Material and Methods section: please add a supportive information/references, why authors decided to use a threshold of 40%. This is important, because choosing the contouring method has an influence on the obtained results, especially in MTV and SUVmean (which a product of TLG).

5.       Results section: Please add an information if patients with distant metastases (M1) had higher or lower or equal values of assessed parameters for primary tumor within the race?

6.       Results section, page 5: does Authors checked the differences within the Black and Mixed ancestry between subtypes and PET-derived parameters? This would be more interested for clinical point of view, than showing that Black race has higher parameters.

7.       Results section, Tables: please add the abbreviation for SD which is used in every table.

8.       Discussion: Please remove Figure 1 from Discussion section – the figure should be placed either in Material and Methods section, or in the Results section.

Round 2

Reviewer 1 Report

Thank you for your response to my questions.
As you mentionned in your introduction, depending on socioeconomic factors, there may differences of the extent of the disease at diagnosis. As there was no mesure of the size of the tumor at diagnosis, and that black patients presented with more distant metastasis than other races, how can you determine that higher SUV, MTV or TLG are due to biological differences among races and not to later detection of a more advanced disease?

You provided very interesting results, although the potential multifactorial causes of differences in SUV parameters among races could be further adressed.

Reviewer 2 Report

Thank you very much for your prompt responses.

The authors performed the changes according to my recommendations, adding the needed explanations.

 I have no further suggestions to make and consider that the manuscript is in a good form for publication. 

Author Response

Thank you

Reviewer 3 Report

no comments

Author Response

Thank you